# Development and Validation of Multiplex-PCR Assay for *β-Carotene hydroxylase* and *γ-Tocopherol methyl transferase* Genes Governing Enhanced Multivitamins in Maize for Its Application in Genomics-Assisted Breeding

**DOI:** 10.3390/plants14010142

**Published:** 2025-01-06

**Authors:** Munegowda Manoj Gowda, Vignesh Muthusamy, Rashmi Chhabra, Hriipulou Duo, Saikat Pal, Nisrita Gain, Ashvinkumar Katral, Ravindra K. Kasana, Rajkumar U. Zunjare, Firoz Hossain

**Affiliations:** Division of Genetics, ICAR-Indian Agricultural Research Institute, New Delhi 110012, India; manojgowdam95@gmail.com (M.M.G.); reshu0428@rediffmail.com (R.C.); hriipuloudui@gmail.com (H.D.); saikatpalag64@gmail.com (S.P.); nisrita.gain@gmail.com (N.G.); agk_gpb@yahoo.com (A.K.); ravindracsa1@gmail.com (R.K.K.); raj_gpb@yahoo.com (R.U.Z.); fh_gpb@yahoo.com (F.H.)

**Keywords:** maize, uniplex- and multiplex-PCR, cost and time, genomics-assisted breeding, biofortification

## Abstract

Traditional maize possesses low concentrations of provitamin-A and vitamin-E, leading to various health concerns. Mutant alleles of *crtRB1* and *vte4* that enhance β-carotene (provitamin-A) and α-tocopherol (vitamin-E), respectively, in maize kernels have been explored in several biofortification programs. For genetic improvement of these target nutrients, uniplex-PCR assays are routinely used in marker-assisted selection. However, due to back-to-back breeding seasons, the time required for genotyping individually for each target gene in large backcross populations becomes a constraint for advancing the generations. Additionally, multiple PCR assays for various genes increase the required costs and resources. Here, we aimed to develop a multiplex-PCR assay to simultaneously identify different allelic forms of *crtRB1* and *vte4* genes and validate them in a backcross-based segregating population. The PCR assay was carried out using newly developed primers for *crtRB1* and a gene-specific primer for *vte4*. The uniplex-PCR assay was standardized for selected primer pairs in the BC_1_F_1_ population segregating for *crtRB1* and *vte4* genes. Subsequently, a multiplex-PCR assay for *crtRB1* and *vte4* genes was developed and employed for genotyping in the BC_1_F_1_ population. The assay differentiated among four possible genotypic classes, namely *crtRB1^+^crtRB1/vte4^+^vte4*, *crtRB1crtRB1/vte4^+^vte4*, *crtRB1^+^crtRB1/vte4^+^vte4^+^*, and *crtRB1crtRB1/vte4^+^vte4^+^.* This newly developed multiplex-PCR assay saved 41.7% of the cost and 35.6% of the time compared to two individual uniplex-PCR assays. The developed assay could accelerate maize nutritional quality breeding programs through rapid and cost-effective genotyping for the target genes. This is the first report of a multiplex-PCR assay specific to *crtRB1* and *vte4* genes for its use in genomics-assisted breeding in maize.

## 1. Introduction

Malnutrition resulting from inadequate consumption of a balanced diet has emerged as one of the most alarming health problems, especially among children and pregnant women [1]. It leads to poor health and increased susceptibility to diseases, contributing to significant socio-economic losses [2,3]. Globally, it is estimated that two billion people are suffering due to malnutrition, and 821 million people are undernourished [4]. Nearly 50% of deaths among children under five years of age are also linked to malnutrition, and these are mostly prevalent in low- and middle-income countries [5].

Among the different micronutrients, the deficiency of vitamin-A (VAD) is quite prevalent and affects nearly 5.2 million preschool-age children and 19 million pregnant women [6,7]. Vitamin-A is essential for proper growth, vision, immunity, and reproductive functions in humans, and it is necessarily supplied through diet [8,9]. Adult non-pregnant and non-lactating women require 500 μg/day of vitamin-A, while its deficiency leads to impaired vision, night blindness, impeded growth, and increased susceptibility to various infectious diseases [10]. Vitamin-E or tocopherol is another essential vitamin required for proper reproductive functions and maintenance of cell integrity [11,12]. It acts as a potential antioxidant and prevents the damaging effects of the free radicals produced during metabolism [13]. Tocopherols also play an important role in preventing Alzheimer’s, cardiovascular, neurological, degenerative, and inflammatory diseases [12,14]. Primarily, vitamin-E consists of α-, β-, γ-, and δ- fractions;among these, γ-tocopherol constitutes 80% of the total fractions [15]. However, α-tocopherol becomes effectively absorbed six times more by the receptors in the body than γ-tocopherol, and, notably, α-tocopherol possesses the highest vitamin-E activity [12,16]. The dietary needs of humans notably include the requirement of 15 mg/day of vitamin-E [17]. However, common diets are deficient in vitamin-E, leading to sub-optimal levels among 20% of the global population [18,19].

Maize is an important cereal that provides 30% of food-calorie intake to more than 4.5 billion people across the world [20,21]. However, the traditional yellow maize varieties possess low concentrations of provitamin-A (0.25–2.50 µg/g dry weight) [22,23,24] and α-tocopherol (6–8 µg/g dry weight) [15,25]. The naturally available mutant version of *β-carotene hydroxylase* (*crtRB1*) significantly increases the level of provitamin-A [26,27,28,29] by restricting the conversion of β-carotene (BC) into β-cryptoxanthin (BCX) and further BCX to zeaxanthin (ZEA), thereby enhancing the accumulation of BC [26,30,31]. The 3′TE polymorphism (insertions of 325 bp or 1250 bp) of the *crtRB1* gene that spans the sixth exon results in unfavorable alleles, whereas the allele without the TE insertion is the favorable allele responsible for enhancing the presence of β-carotene [26,30]. Another natural mutant called *γ-tocopherol methyl transferase* (*vte4*) enhances the accumulation of α-tocopherol by converting γ-tocopherol [18]. Two insertion/deletion (InDel) markers, *InDel7* and *InDel118,* located in 5′UTR and the promoter regions within the *vte4* gene, were identified to be significantly associated with higher levels of α-tocopherol in maize kernels [15,18]. The favorable haplotypes of the *vte4* gene accumulate 3.2-fold more α-tocopherol than the unfavorable haplotype in maize kernels [18]. The introgression of favorable alleles of *crtRB1* and *vte4* genes was reported to accumulate 2- to 10-fold higher provitamin-A [1,7,27,32] and 2- to 4-fold higher vitamin-E [1,7,32] compared to normal genotypes with unfavorable alleles, respectively.

Owing to the greater significance of these nutritional qualities in maize biofortification, stacking of *crtRB1* and *vte4* in a single genetic background holds immense potential in alleviating malnutrition [1,25]. In plant breeding programs, the genotyping of segregating populations for each target gene requires a uniplex-PCR assay, which is expensive, laborious, and time-consuming [29]. Alternatively, a new genotyping tool called the Kompetitive Allele-Specific PCR (KASP) assay is not feasible in common molecular laboratories associated with plant breeders where the intensity of the breeding program is low and moderate population sizes are targeted for genotyping. Further, breeders from developing and underdeveloped countries cannot afford such high costs in genotyping platforms [7,33]. Moreover, the need for sophisticated laboratory conditions, the time and resources associated with genotyping, and the demands of skilled manpower further pose great constraints for the use of the KASP assay by plant breeders.

An innovative approach called multiplex-PCR is a variant of the conventional PCR, where more than one locus or target gene is amplified simultaneously in a single reaction [34]. It provides potential advantages over uniplex-PCR due to fewer reactions being required in genotyping, which reduces the cost, labor, and time required for genotyping several target genes in marker-assisted selection (MAS) programs. Until now, no previous efforts have been undertaken regarding the simultaneous selection of different alleles of *crtRB1* and *vte4* genes through a single multiplex-PCR approach. Thus, the present investigation was designed to (i) develop a multiplex-PCR assay for *crtRB1* and *vte4* genes, (ii) validate the multiplex-PCR assay in a large backcross-based segregating population, and (iii) compare the multiplex-PCR assay with the individual gene-based uniplex-PCR assay regarding its application in genomics-assisted breeding programs for enhancing maize nutritional quality.

## 2. Results

### 2.1. Selection of Functional Marker for crtRB1 Gene

The polymorphism in the *crtRB1* gene is due to the insertion of either the 325 bp or 1250 bp transposable element (TE) that spans the sixth exon and the 3′UTR (untranslated region) of the gene (Figure 1A), which could generate three possible alleles, viz. *allele1* without any insertion, *allele 2* with 325 bp insertion, and *allele 3* with insertion of 1250 bp (Figure 1B; Table 1). Among the three putative primer pairs designed for the *crtRB1* gene, the set 3 primer pair was selected for further analysis as the PCR conditions were compatible for multiplexing with another gene primer pair. Further, this primer pair could differentiate between favorable and unfavorable alleles with the selected marker when validated with the known genotypes (Figure 2). The genotypes with a favorable allele produced an amplicon of 544 bp, and the unfavorable allele generated an amplicon of 213 bp. Another unfavorable amplicon, i.e., 869 bp, was not produced due to the absence of the 325 bp TE insertion in the screened genotypes. Other unfavorable amplicons, namely 1138 bp and 1749 bp, were not produced due to their larger size (>1000 bp) as they require modifications in the PCR conditions.

### 2.2. Uniplex-PCR Assay for crtRB1 Gene

The PCR assay for the *crtRB1* gene was standardized for the newly designed primer pairs. Following the standardized PCR conditions, the uniplex-PCR assay was carried out in the BC_1_F_1_ population. Resolving the amplicons identified two kinds of genotypes, namely homozygous for the favorable allele of *crtRB1* (*crtRB1crtRB1*: lane numbers 1, 4, 8, 10, 11, 12, 17, 18, 20, 21, and 23) and heterozygous for the unfavorable allele of *crtRB1* (*crtRB1*^+^*crtRB1*: lane numbers 2, 3, 5, 6, 7, 9, 13, 14, 15, 16, 19, and 22), which were distinguishable among the genotypes in the segregating population (Figure 2A). Further, the genotyping of 96 samples for the *crtRB1* gene involved an approximate cost of INR 1710.41/USD 21.39, which could be completed in 4.5 h (Table 2 and Table 3).

### 2.3. Uniplex-PCR for vte4 Gene

The genotypes heterozygous for *vte4* (*vte4*^+^*vte4*: lane numbers 1, 5, 6, 8, 9, 11, 13, 15, 16, 17, 19, 20, and 21) and unfavorable homozygous for the *vte4* (*vte4*^+^*vte4*^+^: lane numbers 2, 3, 4, 7, 10, 12, 14, 18, 22, and 23) gene were identified by genotyping in BC_1_F_1_ (Figure 2B). The amplicon size of the favorable allele was 373 bp, while the unfavorable allele produced a 491 bp amplicon (Figure 2B). Further, the genotyping of 96 samples for the *vte4* gene involved an approximate cost of INR 1717.29/USD 21.48 with a time of 4.5 h (Table 2 and Table 3).

### 2.4. Multiplex-PCR Assay for crtRB1 and vte4 Genes

After genotyping the BC_1_F_1_ population with uniplex assays for both the *crtRB1* and *vte4* genes, the multiplex-PCR assay was performed for the simultaneous detection of both favorable and unfavorable alleles of these genes in the same BC_1_F_1_ population (Figure 2C; Table 4). All four possible genotypic classes, (i) homozygous favorable for *crtRB1* (*crtRB1crtRB1*) and heterozygous for *vte4* (*vte4*^+^*vte4*) (lane numbers 1, 8, 11, 17, 20, and 21), (ii) homozygous favorable for *crtRB1* (*crtRB1crtRB1*) and homozygous unfavorable for *vte4* (*vte4^+^/vte4^+^*) (lane numbers 4, 10, 12, 18, and 23), and (iii) heterozygous for *crtRB1* (*crtRB1^+^crtRB1*) and homozygous unfavorable allele for *vte4* (*vte4^+^vte4^+^*) (lane numbers 2, 3, 7, 14, and 22) and heterozygous for *crtRB1* (*crtRB1^+^crtRB1*) and homozygous unfavorable allele for *vte4* (*vte4^+^vte4^+^*) (lane numbers 5, 6, 9, 13, 15, 16, and 19), were identified. The calculated cost and required time for genotyping of 96 samples in a multiplex-PCR assay were INR 1997.56 and 5.8 h, respectively. Thus, there was a 41.7% reduction in the cost and a 35.6% reduction in the time compared to the total cost and time involved in two uniplex-PCR assays (Table 2 and Table 3).

## 3. Discussion

Maize is widely used for various industrial purposes, including food, feed, and bioenergy production worldwide [35]. However, the deficiency of essential vitamins, such as vitamin-A and vitamin-E, in maize-based diets has contributed to significant malnutrition, particularly in developing and underdeveloped countries [4]. The available gene-based markers and PCR protocols of *crtRB1* and *vte4* genes were routinely used for individual gene-based selections in genomics-assisted breeding programs [1,7]. Despite their effectiveness, the routine approach of selecting individual genes in separate assays is time- and resource-intensive, especially given the limited time between breeding seasons and the constraints on genotyping resources [36]. This investigation deals with the development and validation of an innovative multiplex-PCR assay, aimed at accelerating maize biofortification efforts by enabling the simultaneous selection of target alleles. This approach could significantly streamline the genomics-assisted breeding process by combining the assays for multiple genes into a single step.

### 3.1. Functional Polymorphism of crtRB1 and vte4 Genes Governing Enhanced Multivitamins

The *crtRB1* gene, located on chromosome 10, hydroxylates a major amount of β-carotene to produce β-cryptoxanthin (provitamin-A activity, only half of β-carotene) and further zeaxanthin (no provitamin-A activity) [26,31]. The polymorphism in the *crtRB1* gene due to *allele1* (without insertion) is a favorable allele that reduces the activity of the *β-carotene hydroxylase* enzyme and causes a higher accumulation of provitamin-A carotenoids [25]. Likewise, the *vte4* gene catalyzes the conversion of γ-tocopherol to α-tocopherol, and the favorable allele with the deletion of the 118 bp in 5′UTR and the promoter region is known to increase the α-tocopherol concentration by 3.2-fold [12,18].

### 3.2. Reproducibility of Markers for Genotyping crtRB1 and vte4 Genes

The primers reported by Yan et al. [25] for the *crtRB1* gene showed inconsistent and non-reproducible results in subtropical germplasm. Therefore, the present study developed new primer sets that are reproducible and compatible with multiplexing. In a heterozygous genotype with one of the insertions (325 bp or 1250 bp), each primer set could generate four amplicons (in the 325 bp insertion) or five amplicons (in the 1250 bp insertion) (Table 1; Figure 1). To overcome this, the PCR amplification conditions were standardized in such a way that only small-sized amplicons (up to 600 bp) could be amplified. Among the three putative sets of primers for the *crtRB1* gene, primer set 3 was selected for the standardization of the multiplex-PCR assay with the *vte4* gene. In primer set 1, the difference between the favorable (175 bp) and unfavorable amplicons (171 bp) was minimal (4 bp), which would require more time to resolve the amplicons through electrophoresis. In primer in set 2, the amplicon size of the favorable allele of *crtRB1* was 471 bp, which was close to the amplicon size of the unfavorable allele of *vte4* (491 bp). This proximity could cause difficulties in accurately classifying these two amplicons during the multiplex-PCR assay when identifying different allelic classes. Therefore, primer set 3 was selected and validated in known genotypes, where it revealed distinguishable polymorphisms. The PCR conditions and protocol for a uniplex-PCR assay were standardized for newly designed primers using a gradient PCR technique. The gradient PCR was performed with varying concentrations of primers, master mix, and template DNA. In the case of *vte4*, the PCR reaction conditions and amplification protocol for the *InDel* (118 bp) reported by Li et al. [18] were not reproducible in our laboratory, so they were standardized for reproducible results.

### 3.3. Potentiality of Multiplex-PCR Assay

During the stacking of genes for genetic improvement in both provitamin-A and vitamin-E, genotyping using individual/uniplex assays for each target gene is costly, laborious, and time-consuming due to the handling of large segregating generations in a breeding program [36]. In plants, the multiplex-PCR assay is widely used for the simultaneous detection of different pathogens [37] and for identifying genetically modified crops in quarantine via commonly used markers, promoters, and terminator sequences [38,39]. In the present investigation, the multiplex-PCR assay for *crtRB1* and *vte4* genes was standardized considering the concentration of template DNA, gene-specific primers, and PCR master mix. Notably, in the case of multiplex-PCR, a minor increase in the concentration of the reaction mixtures (primers and master mix) compared to the uniplex-PCR assays occurred as there was an increased probability of the formation of primer dimers when more primers were present in a single reaction mixture. Moreover, the amplification of more than one locus also requires additional dNTPs, MgCl_2,_ and *Taq* polymerase.

### 3.4. Effectiveness of Genotyping Cost and Time Between Uniplex- and Multiplex-PCR Assays

The crossing between contrasting maize genotypes, PMI-PV1 (*crtRB1*/*crtRB1*: high in vitamin-A; *vte4^+^/vte4^+^*: low in vitamin-E) and MGU-*vte4*-23 (*crtRB1*^+^/*crtRB1*^+^: low in vitamin-A; *vte4/vte4*: high in vitamin-E) successfully led to the development of the F_1_, which is heterozygous for both the target genes. Further, crossing the F_1_ (*crtRB1*^+^/*crtRB1*/*vte4^+^/vte4*) as a male parent with PMI-PV1 as a recurrent parent (female) resulted in the development of the BC_1_F_1_ population. In this population, four genotypic classes were possible, including (i) *crtRB1*^+^*crtRB1*/*vte4^+^vte4*, (ii) *crtRB1crtRB1*/*vte4^+^vte4*, (iii) *crtRB1*^+^*crtRB1*/*vte4^+^vte4^+^*, and (iv) *crtRB1crtRB1*/*vte4^+^vte4^+^*. The most favorable genotypic class, *crtRB1*/*crtRB1*/*vte4^+^/vte4*, requires selection through markers for further backcrossing. Therefore, two genes were targeted for selection in the entire BC_1_F_1_ population using uniplex-PCR assays specific to each of the *crtRB1* and *vte4* genes. However, adding more target genes increases both cost and time. Subsequently, we calculated the reductions in time and cost by adopting the newly developed multiplex-PCR assay for *crtRB1* and *vte4* genes compared to uniplex-PCR assays. The multiplex-PCR assay developed here saved ~42% of the cost and ~36% of the time when compared with the total cost and time involved in two uniplex-PCR assays. Zunjare et al. [29] developed a multiplex-PCR assay for *crtRB1* and *lcyE* genes governing provitamin-A carotenoids in maize and saved 41% of the cost and 50% of the time as compared to individual uniplex assays. However, the utmost care has to be taken while considering multiplexing for more genes to have similar annealing temperatures. Notably, the amplicon lengths of each allele should be sufficiently different for easier identification of the target gene. However, in the current study, the possible amplicons with different lengths and similar annealing temperatures for both *crtRB1* and *vte4* genes were favored for obtaining the optimum results without any difficulties. Thus, the multiplex-PCR assay developed and validated in the present study would offer immense potential in maize biofortification programs aimed at combining multiple nutrients in the same genetic background. This is the first report regarding the development of a multiplex-PCR protocol for *crtRB1* and *vte4* genes in maize for simultaneous determination of the allelic status of the *crtRB1* and *vte4* genes rapidly and cost-effectively.

## 4. Materials and Methods

### 4.1. Development of Segregating Population for crtRB1 and vte4 Genes

Two biofortified inbreds, viz. PMI-PV1(*crtRB1crtRB1*/*vte4^+^vte4^+^*) and MGU-*vte4*-23 (*crtRB1*^+^*crtRB1^+^*/*vte4vte4*), developed at ICAR—Indian Agricultural Research Institute (IARI), New Delhi, were crossed during the rainy season of 2017 at IARI, New Delhi (28°08′ N, 77°12′ E, 229 MSL). The wild-type/unfavorable alleles were designated as *crtRB1*^+^ and *vte4^+^*, while the mutant alleles/favorable alleles were designated as *crtRB1* and *vte4*. The F_1_s were raised at ICAR—Indian Institute of Maize Research, Winter Nursery Centre, Hyderabad (17°36′ N, 78°47′ E, 536 MSL) during the winter season of 2017–18. F_1_s were tested for hybridity using the gene-based markers of *crtRB1* and *vte4*. The true F_1_s were crossed as male with the PMI-PV1 as the recurrent parent. The BC_1_F_1_ population having 141 genotypes was raised in ICAR-IARI, New Delhi, during the rainy season of 2018. Randomly selected 96 BC_1_F_1_ genotypes were used for standardizing the uniplex-PCR and subsequently multiplex-PCR assays for both *crtRB1* and *vte4* genes.

### 4.2. Functional Markers for crtRB1 and vte4 Genes

Based on sequence information of *crtRB1* gene with (i) insertion of transposable elements (TE: 325 bp and 1250 bp) in the 3′UTR (GQ889716.1) (wild-type: unfavorable allele) and (ii) without insertion (GQ889870.1) (mutant-type: favorable allele), three sets of primers were designed in the study (Table 1; Figure 1) using BatchPrimer3 [40]. Newly designed primers were validated in known genotypes and analyzed for reproducibility in local germplasm. In the case of the *vte4* gene, the information is based on functional polymorphism in the promoter region as reported by Li et al. [18]. Here, we used the *InDel118* marker significantly associated with α-tocopherol in the multiplex assay due to the larger size of InDel to avoid overlapping amplicons with better resolution (Table 1).

### 4.3. Primer Synthesis and DNA Isolation for Genotyping

Primers were custom-synthesized by M/s Macrogen (Europe), and the dilutions were created for a final concentration of 10 μM using Milli-Q water. The extraction of genomic DNA in the BC_1_F_1_ population was carried out by following a standard CTAB method [41], and the isolated DNA was stored in a −20 °C refrigerator.

### 4.4. Uniplex-PCR Assay for crtRB1 and vte4 Genes

The PCR protocol for *crtRB1* of newly designed primers was standardized using different concentrations of primers, Taq polymerase, and template DNA. For amplification of the *crtRB1* gene, uniplex-PCR was carried out in a 25 μL reaction volume consisting of 100 ng template DNA, 1× TaqPlus Polymerase master mix (2X PCR master mix; GBiosciences, St. Louis, MO, USA), 0.2 μM of forward primer, and both the reverse primers (R1 and R2). In the case of amplification of the *vte4* gene, uniplex-PCR was undertaken in a 25 μL reaction having 100 ng template DNA, 1X TaqPlus Polymerase master mix, and 0.4 μM of both forward and reverse primers. Several master mixes were used for analysis, and ‘TaqPlus Polymerase’ master mix provided robust and reproducible results in both uniplex and multiplex assays, which were used in final genotyping. Uniplex-PCR for both *crtRB1* and *vte4* genes was carried out using a GenePro thermal cycler with the PCR conditions as follows: (i) initial denaturation at 95 °C for 5 min, (ii) 35 cycles were run with denaturation (95 °C for 45 s), primer annealing (60 °C for 45 s), primer extension (72 °C for 45 s), and (iii) final extension at 72 °C for 5 min.

### 4.5. Multiplex-PCR Assay

In the multiplex-PCR assay, the concentration of template DNA, primers, and necessary reagents was increased and optimized based on the number of products to be amplified. Here, the optimization was done with varying concentrations of template DNA, primers, master mix, and different PCR conditions. The final multiplex-PCR assay was performed with a reaction volume of 25 μL with 150 ng of template DNA, 1.25X TaqPlus Polymerase master mix, 0.2 μM of forward and reverse primers for *crtRB1*, and 0.5 μM of both forward and reverse primers for *vte4* gene. The amplification conditions for multiplex-PCR included (i) initial denaturation at 95 °C for 10 min, (ii) 40 cycles of denaturation (95 °C for 50 s), primer annealing (60 °C for 50 s), primer extension (72 °C for 50 s), and (iii) final extension at 72 °C for 6 min.

### 4.6. Resolving and Visualization of Amplicons

The amplified products of uniplex- and multiplex assays were separated by running on 2% Seakem LE Agarose gel (Lonza, Rockland, ME, USA) at 100 V for 3–4 h. The banding pattern was visualized through a gel documentation system, and the amplicons were scored for the favorable and/or unfavorable allelic status of *crtRB1* and *vte4* genes in the BC_1_F_1_ backcross population.

### 4.7. Analysis of Cost and Time of Genotyping

The cost and time of genotyping involved in the selection of *crtRB1* and *vte4* genes were calculated for both uniplex- and multiplex assays. Genotyping cost was computed based on the cost of the chemicals, consumables, and plastic wares involved in the assay. The percent reduction in the cost and time was calculated as the ratio of the difference between cost/time incurred in two different uniplex assays (*crtRB1* and *vte4*) and multiplex assay divided by the sum cost/time incurred in two individual uniplex assays.

## 5. Conclusions

Marker-assisted stacking of *crtRB1* and *vte4* genes is routinely achieved through uniplex-PCR assays. The newly developed multiplex-PCR assay effectively differentiated various allelic forms of *crtRB1* and *vte4* genes in a segregating population. This innovative multiplex-PCR assay approach, which enables the simultaneous detection of target genes governing multivitamins, saved 35.6% of the time and 41.7% of the resource cost compared to uniplex-PCR assays for individual genes. This approach could significantly streamline efforts to improve maize nutritional quality through genomics-assisted breeding by combining assays for multiple genes into a single step.

## Figures and Tables

**Figure 1 plants-14-00142-f001:**
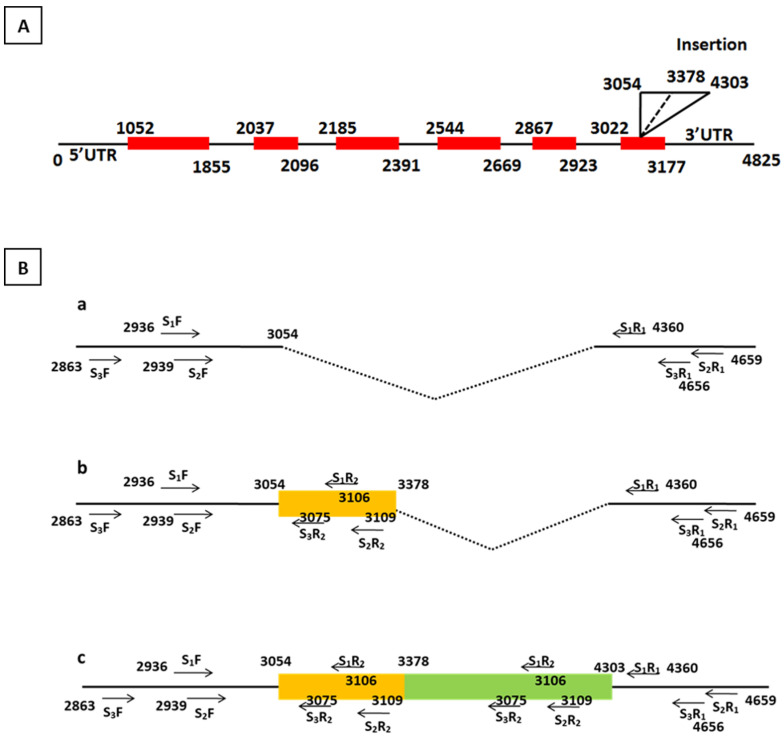
(**A**) Schematic diagram of *crtRB1* gene; (**B**) schematic representation of 3′TE polymorphisms in *crtRB1* gene and primer binding site of a different set of primers. (**a**) *Allele 1* (without insertion), (**b**) *allele 2* (insertion of 325 bp), and (**c**) *allele 3* (insertion of 1250 bp). Numbers in the illustration are based on the GQ889716 sequence.

**Figure 2 plants-14-00142-f002:**
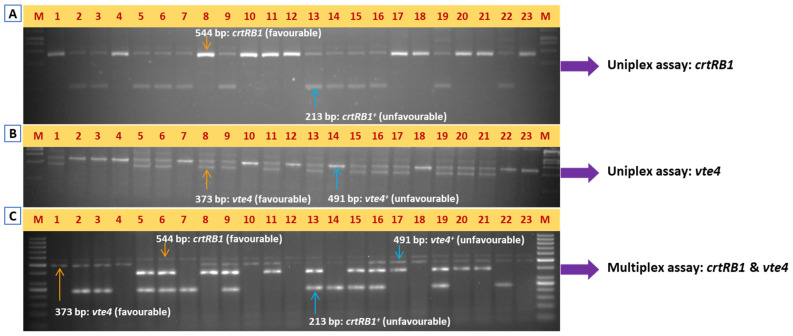
Genotyping of BC_1_F_1_ population segregating for favorable and unfavorable alleles of *crtRB1* and *vte4*. (**A**) Genotyping for *crtRB1* by uniplex-PCR; (**B**) genotyping for *vte4* by uniplex-PCR; (**C**) genotyping for *crtRB1* and *vte4* by multiplex-PCR. M represents 50 bp ladder; 1–23 represents the BC_1_F_1_ plants segregating for both *crtRB1* and *vte4* genes.

**Table 1 plants-14-00142-t001:** Primer sequence information of *crtRB1* and *vte4* genes used in the study.

Gene	Primer Set	Primer Name	Sequence (5′-3′)	Amplicon Size and Condition		Reference
				Favorable Allele(s)	Unfavorable Allele(s)	
*crtRB1*	Set 1	S_1_F	F: TGTGGCCCTTCTTCTTTTGT	175 bp (S_1_F-S_1_R_1_: no insertion)	171 bp (S_1_F-S_1_R_2_: 325 bp or 1250 bp insertion), 500 bp (S_1_F-S_1_R_1_: 325 bp insertion), 1096 bp (S_1_F-S_1_R_2_: 1250 bp insertion), and 1425 bp (S_1_F-S_1_R_1_: 1250 bp insertion)	Designed in the present study
		S_1_R_1_	R1: CAGCTCCATGTACGAATCCA			
		S_1_R_2_	R2: TTTCTTCTCCTGCGTCTTCA			
	Set 2	S_2_F	F: GGCCCTTCTTCTTTTGTCCT	471 bp (S_2_F-S_2_R_1_: no insertion)	171 bp (S_2_F-S_2_R_2_: 325 bp or 1250 bp insertion), 796 bp (S_2_F-S_2_R_1_: 325 bp insertion), 1096 bp (S_2_F-S_2_R_2_: 1250 bp insertion), and 1721 bp (S_2_F-S_2_R_1_: 1250 bp insertion)	
		S_2_R_1_	R1: AACAGCAATACAGGGGACCA			
		S_2_R_2_	R2: TCTTTTCTTCTCCTGCGTCTTC			
	Set 3	S_3_F	F: GCAGATACACCACATGGACAA	544 bp (S_3_F-S_3_R_1_: no insertion)	213 bp (S_3_F-S_3_R_2_: 325 bp or 1250 bp insertion), 869 bp (S_3_F-S_3_R_1_: 325 bp insertion), 1138 bp (S_3_F-S_3_R_2_: 1250 bp insertion), and 1749 bp (S_3_F-S_3_R_1_: 1250 bp insertion)	
		S_3_R_1_	R1: AGCAATACAGGGGACCAGAA			
		S_3_R_2_	R2: TTCACTCACCGGACTGCTAA			
*vte4*	*InDel118*	InDel118_F	F: AAAGCACTTACATCATGGGAAAC	373 bp	491 bp	Li et al. [18]
		InDel118_R	R: TTGGTGTAGCTCCGATTTGG			

F: forward primer; R1: reverse primer 1; R2: reverse primer 2; bp: base pair.

**Table 2 plants-14-00142-t002:** Comparison of cost (INR and USD) for uniplex- and multiplex-PCR assays.

S. No.	Activity	Uniplex-PCR for *crtRB1*		Uniplex-PCR for *vte4*		Multiplex-PCR for *crtRB1* + *vte4* (INR)	
		INR	USD	INR	USD	INR	USD
1.	DNA isolation	13.00	0.16	13.00	0.16	13.00	0.16
2.	Primer synthesis	16.512	0.21	23.39	0.29	44.66	0.56
3.	PCR plate	231.80	2.90	231.80	2.90	231.80	2.90
4.	Master Mix	1056.00	13.21	1056.00	13.21	1315.00	16.45
5.	Micro tips	93.10	1.16	93.10	1.16	93.10	1.16
6.	Gel electrophoresis	300	3.75	300	3.75	300	3.75
7.	Sub-total	1710.41	21.39	1717.29	21.48	1997.56	24.99
8.	Total cost	3427.70 INR/42.87 USD				1997.56	24.99
**Percent (%) reduction**						**41.72**	

INR: Indian Rupee; USD: United States Dollar.

**Table 3 plants-14-00142-t003:** Comparison of time for uniplex- and multiplex-PCR assays.

S. No.	Activity	Uniplex-PCR for *crtRB1* (Hour)	Uniplex-PCR for *vte4* (Hour)	Multiplex-PCR for *crtRB1* + *vte4* (Hour)
1.	Preparation of PCR	0.5	0.5	0.5
2.	Completion of PCR run	2.0	2.0	2.3
3.	Completion of gel electrophoresis	2.0	2.0	3.0
4.	Sub-total	4.5	4.5	5.8
5.	Total time	9.0		5.8
**Percent (%) reduction**				**35.56**

**Table 4 plants-14-00142-t004:** Segregation patterns of *crtRB1* and *vte4* genes in BC_1_F_1_ generation using uniplex- and multiplex assays.

Parentage	N	Uniplex		Multiplex	
*crtRB1*					
PMI-PV1 (*crtRB1crtRB1* & *vte4^+^vte4^+^*) × MGU-vte4-23 (*crtRB1^+^crtRB1^+^* & *vte4vte4*)	141	*CC*	*Cc*	*CC*	*Cc*
		*82*	*59*	*82*	*59*
*vte4*					
PMI-PV1 (*crtRB1crtRB1* & *vte4^+^vte4^+^*) × MGU-vte4-23 (*crtRB1^+^crtRB1^+^* & *vte4vte4*)	141	*VV*	*Vv*	*VV*	*Vv*
		*77*	*64*	*77*	*64*

N: number of plants; *CC: crtRB1^+^crtRB1^+^*; *Cc: crtRB1^+^crtRB1*; *VV: vte4^+^vte4^+^*; *Vv: vte4^+^vte4*. Allele with ‘+’ sign indicates the wild-type (unfavorable allele), and the allele without any sign indicates the mutant (favorable allele).

## Data Availability

All the relevant data are presented in the manuscript. Further inquiries can be directed to the corresponding author.

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
