# Peer review of "Development and Validation of Multiplex-PCR Assay for *β-Carotene hydroxylase* and *γ-Tocopherol methyl transferase* Genes Governing Enhanced Multivitamins in Maize for Its Application in Genomics-Assisted Breeding"

_plants, 2025, doi:10.3390/plants14010142_

Round 1

Reviewer 1 Report (New Reviewer)

Comments and Suggestions for Authors

The paper represents a commendable effort to construct a multiplex-PCR assay for crtRB1 and vte4 genes for its application in maize MAS breeding program. This is an essential contribution to efforts to improve the content of vitamin-A and vitamin-E in maize kernels to alleviate micronutrient malnutrition. Such useful information should be presented in as comprehensible a manner as possible. The highlights of this manuscript are the table 1 and 2 with concrete and reliable comparison the cost and time among different methods. However, there are a number of problems with the text that should be addressed.

Minor Essential Revisions

1. The words “corn” and maize mixed to use, please uniform to one expression.

2. Too much keywords, please reduce to about five keywords.

3. Some data are overage such as the reference [5], the information is 2013, please renew the newly data and reference.

4. Figure 1, the DNA fragments of vte4 genes are faint, please further optimized the PCR condition to get reliable results for utilization in maize breeding program.

5. Figure 1, please interchange the order of figure 1A and 1B to make the logic of figures be consistent with the context that crtRB1 in the front and vte4 in the later.

6. The words in the table-head should be the singular form instead of plural form.

7. All tables should change to three-line table.

8. The title of 3.1 is inappropriate for without cohesion in context, it may change to “Functional polymorphism of crtRB1 and vte4 genes governing enhanced multivitamins”.

9. Materials and Methods: please interchange the parts of 4.1 and 4.2 to make the logic of text closer.

10. Materials and Methods: 4.5 the 1.25X TaqPlus Polymerase master mix may be not appropriate for multiplex-PCR, it’s recommend to use High-Fidelity DNA polymerase instead of master mix to get more clear DNA fragments for vte4 genes.

11. Please uniform the format of references: some reference with issue number some without; some titles of are only capital initial letter of sentence, some are capital letter of each word such as reference No. 4.

Author Response

Response to reviewers’ comments

*Note: Lines numbers provided in the response to each comment here are according to the track change copy of the manuscript.

Reviewer #1

Comment 1: The paper represents a commendable effort to construct a multiplex-PCR assay for crtRB1 and vte4 genes for its application in maize MAS breeding program. This is an essential contribution to efforts to improve the content of vitamin-A and vitamin-E in maize kernels to alleviate micronutrient malnutrition. Such useful information should be presented in as comprehensible a manner as possible. The highlights of this manuscript are the table 1 and 2 with concrete and reliable comparison the cost and time among different methods. However, there are a number of problems with the text that should be addressed.

Response 1: Thank you very much for recognizing the essence of the work, advancements, and significance of the current study. We greatly appreciate your kind efforts and time for the critical assessment and valuable insights for further improvement of the manuscript.

Comment 2: The words “corn” and maize mixed to use, please uniform to one expression.

Response 2: Thanks for the suggestion. We used the word ‘maize’ to have uniform expression throughout the manuscript (Line 30).    

Comment 3: Too much keywords, please reduce to about five keywords.

Response 3: We have restricted the keywords to five numbers, as per the suggestion (Lines 30-31).  

Comment 4: Some data are overage such as the reference [5], the information is 2013, please renew the newly data and reference.

Response 4: Thank you very much for the worthy suggestion. We have updated the latest data with supporting literature (WHO 2024) (Lines 38-40).   

Comment 5: Figure 1, the DNA fragments of vte4 genes are faint, please further optimized the PCR condition to get reliable results for utilization in maize breeding program.

Response 5: We appreciate your concern regarding the clarity of the amplicons. We have revised the Figure 1 (revised Figure 2), as per the suggestion for better clarity (Lines 138-139).

Comment 6: Figure 1, please interchange the order of figure 1A and 1B to make the logic of figures be consistent with the context that crtRB1 in the front and vte4 in the later.

Response 6: We appreciate your efforts in the impactful shaping of the manuscript. We have revised the Figure 1 (revised Figure 2) to represent the amplification pattern in correct, as per the suggestion (Lines 138-139).

Comment 7: The words in the table-head should be the singular form instead of plural form.

Response 7: We have the table-head to singular form, as per the suggestion (Tables 1, 2, 3 & 4).

Comment 8: All tables should change to three-line table.

Response 8: Thank you very much for the suggestion. We revised all the tables to three-line table, as per the suggestion (Tables 1, 2, 3 & 4) (Lines 134-135, 176-199, 226-230).

Comment 9: The title of 3.1 is inappropriate for without cohesion in context, it may change to “Functional polymorphism of crtRB1 and vte4 genes governing enhanced multivitamins”.

Response 9: Thank you very much for your insightful remark. We have revised the section 3.1 title to “Functional polymorphism of crtRB1 and vte4 genes governing enhanced multivitamins”, as per the suggestion (Line 249).

Comment 10: Materials and Methods: please interchange the parts of 4.1 and 4.2 to make the logic of text closer.

Response 10: Thank you greatly for your constructive remarks on improving the manuscript. We have interchanged the sections 4.1 and 4.2 in Materials and Methods, as per the suggestion (Lines 342-365).

Comment 11: Materials and Methods: 4.5 the 1.25X TaqPlus Polymerase master mix may be not appropriate for multiplex-PCR, it’s recommend to use High-Fidelity DNA polymerase instead of master mix to get more clear DNA fragments for vte4 genes.

Response 11: We appreciate your concern regarding the reliability of the findings and their suitability. We could obtain clearer amplification when we used 1.25X TaqPlus Polymerase master mix and couldn’t had any issues with PCR-amplification and clarity of the amplicons. In addition, the similar concentrations of 1.25X TaqPlus Polymerase master mix has been used in earlier reports (Zunjare et al. 2018 Journal of Plant Biochemistry and Biotechnology  https://doi.org/10.1007/s13562-017-0432-8; Baveja et al. 2021 Journal of Plant Biochemistry and Biotechnology  https://doi.org/10.1007/s13562-020-00585-6).

Comment 12: Please uniform the format of references: some reference with issue number some without; some titles of are only capital initial letter of sentence, some are capital letter of each word such as reference No. 4.

Response 12: We have reformatted all the references uniformly according to your suggestions and the journal guidelines. Thank you for your valuable feedback (Lines 452-574).

Reviewer 2 Report (New Reviewer)

Comments and Suggestions for Authors

The reviewed manuscript is dedicated to design and validation of a novel multiplex PCR assay for genotyping of maize in order to select favorable genetic composition with the increased levels of vitamins A and E. The topic itself is timely and interesting and the suggested approach could become a valuable tool for further selection. However, several questions need to be answered before possible publication.

1.      Authors are requested to change the Figure 2 in a way to clearer demonstrate possible amplicons length. Also, criteria for primer selection should be detailed. In this light the section 3.1 would be helpful if transferred in the Results.

2.      The quality of Figure 1 is low, and it would be better to be replaced by gels with fewer samples with only typical genotypes that would be specified in the figure’s legend. Current Figure 1 contains too many lanes which are hard to distinguish. Main marker length also need to be provided in the figure.

3.      Authors stated that the presented multiplex PCR was optimized for the better performance, but did not provided the supporting data.

4.      3.1. Functional polymorphism of crtRB1 and vte4 genes the section seems to be redundant, because this information was given in the Introduction.

5.      Authors are requested to discuss possible limitations of their study.

Author Response

Response to reviewers’ comments

*Note: Lines numbers provided in the response to each comment here are according to the track change copy of the manuscript.

Reviewer #2

Comment 1: The reviewed manuscript is dedicated to design and validation of a novel multiplex PCR assay for genotyping of maize in order to select favorable genetic composition with the increased levels of vitamins A and E. The topic itself is timely and interesting and the suggested approach could become a valuable tool for further selection. However, several questions need to be answered before possible publication.

Response 1: Thank you very much for recognizing the essence of the work, advancements, and significance of the current study. We greatly appreciate your kind efforts and time for the critical assessment and valuable insights for further improvement of the manuscript.

Comment 2: Authors are requested to change the Figure 2 in a way to clearer demonstrate possible amplicons length. Also, criteria for primer selection should be detailed. In this light the section 3.1 would be helpful if transferred in the Results.

Response 2: We greatly appreciate your efforts in shaping the manuscript to enhance its impact. The detailed gene information in Figure 2A (revised Figure number 1A) and the possible primer combinations with their corresponding amplicon lengths in Figure 2B (revised Figure number 1B) are interconnected. Additionally, the possible primer combinations and the amplicon lengths for each combination are clearly presented in Table 4 (revised Table number 1). In line with the suggestion, the part of Section 3.1 has been moved to the Results section 2.1 (Lines 105-109). 

Comment 3: The quality of Figure 1 is low, and it would be better to be replaced by gels with fewer samples with only typical genotypes that would be specified in the figure’s legend. Current Figure 1 contains too many lanes which are hard to distinguish. Main marker length also need to be provided in the figure.

Response 3:  Thank you greatly for suggesting critically. We have revised Figure 1 (revised Figure number 2) to include a high-quality gel image with clearer labels and appropriately marked amplicons. Regarding the fewer samples in the gel image, the previous reviewer in the last review process recommended using a gel image with a larger number of samples. Accordingly, we have provided a gel image with more samples, which better represents all possible combinations of favourable and unfavourable alleles. The manuscript text has also been revised in line with this suggestion (Lines 105-118).

Comment 4: Authors stated that the presented multiplex PCR was optimized for the better performance, but did not provided the supporting data.

Response 4: We appreciate your concern regarding the optimization of PCR conditions. We wanted to highlight here that we used different concentrations of primers, Taq Polymerase master mix, and template DNA and annealing temperature, which are crucial for obtaining optimum amplicon length with better resolution. The optimization of PCR conditions is the basic step in all molecular biology laboratories to fix the conditions as per the focus of the study. This background work seems not required to be mentioned here as the optimized conditions have been well explained in the manuscript. These points have been highlighted with more details in the revised manuscript (Lines 388-414).          

Comment 5: Functional polymorphism of crtRB1 and vte4 genes the section seems to be redundant, because this information was given in the Introduction.

Response 5: Thanks for the worthy suggestion. We have revised this section to make it more comprehensive by avoiding repetition of the content. Additionally, part of this section has been moved to the Results section, as per your earlier suggestion (Lines 105-109, 249-260).   

Comment 6: Authors are requested to discuss possible limitations of their study.

Response 6: Thank you very much for the insightful suggestion. We have discussed the possible limitations of the study in the last paragraph of the discussion section, as per the suggestion (Lines 330-335).

Round 2

Reviewer 2 Report (New Reviewer)

Comments and Suggestions for Authors

Many thanks to authors for their thoughtful replies and careful editing the manuscript. All mentioned questions have been cleared and no further changes are needed for publication.

This manuscript is a resubmission of an earlier submission. The following is a list of the peer review reports and author responses from that submission.

Round 1

Reviewer 1 Report

Comments and Suggestions for Authors

The authors developed and validated multiplex-PCR assay for β-carotene hydroxylase and γ-tocopherol methyl transferase genes governing enhanced multivitamins in maize for its application in genomics-assisted breeding, the investigation is meaningful, but there are some inadequacies.

in results 2.4. the authors should provide the figure of  more samples Genotyped for vte4 and crtRB1 by multiplex PCR, but not just 23 samples of BC1F1 population.

in conclusion, "saved 50% of the time and 41% of the resource cost compared to uniplex-PCR assays for individual genes." is different from the results in the abstract and the results part.

Author Response

Reviewer #1:

Comment 1: The authors developed and validated multiplex-PCR assay for β-carotene hydroxylase and γ-tocopherol methyl transferase genes governing enhanced multivitamins in maize for its application in genomics-assisted breeding, the investigation is meaningful, but there are some inadequacies.

Response 1: Thank you greatly for your kind appreciation of our efforts and suggestions for further improvements to the manuscript.

Comment 2: In results 2.4. the authors should provide the figure of more samples Genotyped for vte4 and crtRB1 by multiplex PCR, but not just 23 samples of BC1F1 population.

Response 2: We appreciate the reviewer's efforts in impactful shaping the manuscript. The study involved 141 BC1F1 plants, which were genotyped using both the uniplex and multiplex assays. The obtained results for crtRB1 and vte4 were well-agreed across assays.

Further, we considered randomly selected 96 plants of 141 plants for standardizing the uniplex and multiplex assays for a better understanding of the effectiveness of the developed assays in terms of cost and time. Due to the available genotyping system at our laboratory, at a particular time, we can carry out the genotyping of 23 genotypes apart from a ladder using one comb/line in the gel electrophoresis system. Segregation patterns comparison between the uniplex and multiplex assays have been presented in Table 3. Accordingly, the representative gel picture for the uniplex assay and multiplex assay have been presented in Figure 1.

Comment 3: In conclusion, "saved 50% of the time and 41% of the resource cost compared to uniplex-PCR assays for individual genes." is different from the results in the abstract and the results part.

Response 3: Thank you very much for the critical observation. The values have been crosschecked and corrected for accuracy across the sections as per the suggestion (Lines 24-25, 419-421).

*Note: Line numbers provided in the response to each comment here are according to the track change copy of the manuscript.

Reviewer 2 Report

Comments and Suggestions for Authors

The manuscript brings an example of the effectivity of a PCR Multiplex application in a relevant subject - improvement of the maize nutritional quality  - and should be accepted for publication.

I have just one observation: Table 2; line 157 should be “Total time” instead of “Total cost”.

Author Response

Reviewer #2:

Comment 1: The manuscript brings an example of the effectivity of a PCR Multiplex application in a relevant subject - improvement of the maize nutritional quality - and should be accepted for publication.

Response 1: Thank you greatly for recognizing the essence of the work, advancements, and significance of the current study. We sincerely appreciate your kind efforts and time in reviewing our manuscript.

Comment 2: I have just one observation: Table 2; line 157 should be “Total time” instead of “Total cost”.

Response 2: Thanks for the critical observation. We have revised it to “Total time” as per the suggestion (Lines 173-184).

 *Note: Line numbers provided in the response to each comment here are according to the track change copy of the manuscript.

Reviewer 3 Report

Comments and Suggestions for Authors

Multiplex-PCR assay is a basic bench work in a molecular biology related lab. The multiplex-PCR assay reported in this research could be used as a lab protocol for Vt-E related phenotyping/genotyping by SNP based PCR, such as KASP. The weight and quality of the study are not good enough for publication in MDPI journal.

Comments on the Quality of English Language

Normal

Author Response

Reviewer #3:

Comment 1: Multiplex-PCR assay is a basic bench work in a molecular biology related lab. The multiplex-PCR assay reported in this research could be used as a lab protocol for Vt-E related phenotyping/genotyping by SNP based PCR, such as KASP. The weight and quality of the study are not good enough for publication in MDPI journal.

Response 1: We thank and appreciate the concern of the reviewer regarding the Multiplex-PCR analysis as a basic bench work. We agree that the SNP-based PCR such as KASP would also be employed for genotyping. However, the KASP assay wouldn’t be feasible for genotyping large set of segregating populations in common molecular laboratories associated with the plant breeders as sequence information is a must for KASP assay and breeders of developing and under-developed countries cannot afford to use such high costs in genotyping using NGS platforms. Moreover, the need for sophisticated laboratory conditions, and resources associated with genotyping, and the demands of skilled manpower further pose great constraints for the use of KASP assay by plant breeders.

The multiplex-PCR assay developed in the current study possesses great potential for its use in rapid and cost-effective genotyping in common molecular laboratories by Plant Breeders for advancing the generations with available short-time duration due to back-to-back generations in genomics-assisted breeding programs. These points have been discussed in the introduction section of the revised manuscript (Lines 81-87). 

*Note: Line numbers provided in the response to each comment here are according to the track change copy of the manuscript.

Reviewer 4 Report

Comments and Suggestions for Authors

The aim of the study was to develop a multiplex PCR assay for detection of alleles of two genes, which contribute to higher concentrations of beta-carotene and gamma tocopherol, crtRB1 and vte4 respectively. Based on the reported results, the goal was successfully achieved. However, I would suggest some changes/corrections to the manuscript to improve clarity.

L42: I think a website could be cited in the literature section. It would be better if a specific article would be identified with these reports.

Introduction section: additional information about the mutant alleles of two genes could be included in the introduction. Is there a study with a functional characterisation of these mutations? How are the mutations related to the increase in beta-carotene and gamma tocopherol? Two different insertions are mentioned in Material and Methods (325 bp and 1250 bp). This should be better described in the introduction. You should also indicate when these mutations were identified, whether they were used in some breeding programmes, how they were detected by previous methods, etc.

I will continue with materials and methods before moving on to the results and discussion.

L314: Do you have any additional explanation as to why GQ889716? For example, I see that the sequence is identical to the RefSeq maize genome (GCF_902167145.1). The sequence of the gene according to Gene database is NC_050105.1:c138690462-138687456 with a length of 3006 nt, while 4825 is the length of the sequence stored with GQ889716 (the region with the gene is identical, so it's fine, but I had to check what you selected as no additional information about the selected sequence was included).

L315: GQ889870 is a sequence with 1250 insertion. It is not clear where the information about the 325 bp insertion comes from.

L319: Li et al. have published primers labelled as InDel7. Why did you choose InDel118?

L333: Please add a column with the primer names in the table. For example, Set 1 Forward = S1F. In this way, the names of the primers in Figure 2 are easy to understand. Also indicate which allele is amplified with Reverse-1 and which with Reverse-2. For each unfavourable allele, also indicate which primers are used to amplify it. What is the reason for obtaining unfavourable alleles of 171, 500, 1096 and 1425 bp? For example, for Set-3, only allele 213 bp is mentioned as unfavourable in the Results section, while also alleles 869, 1138 and 1749 bp are mentioned here.

L3712: …reagents was increased. – …were increased.

L401: In the Discussion section it is mentioned that: “The multiplex- PCR assay developed here saved ~42% of the cost and ~36% of the time when compared with the total cost and time involved in two uniplex- PCR assays.” But here 50% of the time and 41%. Which information is correct?

Results

L85: You do not need to insert nucleotide sequences here if they are already contained in Table 3. How was set-3 selected? What does it mean that "it was compatible”? (I saw that this is explained in the Discussion, but it should be here.)

L85: …with aother – …with another

Discussion

L200-L211 – this should be included in the introduction.

L241: Please add in the title that the numbers are based on the sequence GQ889716. Also, I think Figure 2 should be included in the Results section as a primer map and a comparison of primer positions from previous articles could also be provided. Also the resolution of Figure 2 should be increased.

L246: Please include this information in the introduction to justify why you designed new primers.

L247-L265: I think this should be under the Results section. Here you explain well why Set-3 was chosen, but this should be in the results. The Discussion section is where you compare the results to other studies, how your results improved on previous methods, how your methods differ, etc.

L269-L285: This paragraph could be shortened as the general introduction to the multiplex assay is already included in the Introduction section.

L292: Shouldn’t it be PMI-PV1 instead of PMI-PV5?

L296: Why crtRB1/crtRB1/vte4+/vte4 is the most favourable genotype for further backcrossing? Why not homozygotes with vte4 (allele for high content of gamma tocopherol)?

Author Response

Reviewer #4:

Comment 1: The aim of the study was to develop a multiplex PCR assay for detection of alleles of two genes, which contribute to higher concentrations of beta-carotene and gamma tocopherol, crtRB1 and vte4 respectively. Based on the reported results, the goal was successfully achieved. However, I would suggest some changes/corrections to the manuscript to improve clarity.

Response 1: We thank and appreciate your kind efforts and time in reviewing our manuscript with constructive suggestions for further improvement.  

Comment 2: L42: I think a website could be cited in the literature section. It would be better if a specific article would be identified with these reports.

Response 2: We have cited the relevant literature as per the suggestion (Line 42).

Comment 3: Introduction section: additional information about the mutant alleles of two genes could be included in the introduction. Is there a study with a functional characterisation of these mutations? How are the mutations related to the increase in beta-carotene and gamma tocopherol? Two different insertions are mentioned in Material and Methods (325 bp and 1250 bp). This should be better described in the introduction. You should also indicate when these mutations were identified, whether they were used in some breeding programmes, how they were detected by previous methods, etc.

Response 3: Thank you very much for your worthy suggestion. The information regarding the mutant alleles, the mechanism underlying the enhancement of the provitamin-A and vitamin-E, functional characterization, and introgression of these mutant regions in the popular maize genotypes have been included in the introduction section as per the suggestion (Lines 58-76).   

Comment 4: I will continue with materials and methods before moving on to the results and discussion. L314: Do you have any additional explanation as to why GQ889716? For example, I see that the sequence is identical to the RefSeq maize genome (GCF_902167145.1). The sequence of the gene according to Gene database is NC_050105.1:c138690462-138687456 with a length of 3006 nt, while 4825 is the length of the sequence stored with GQ889716 (the region with the gene is identical, so it's fine, but I had to check what you selected as no additional information about the selected sequence was included).

Response 4: The sequence of GQ889716 was utilized in the current study as it was deposited by Yan et al. (2010) from where the first report of crtRB1 came from. The GQ889716 is Zea mays cultivar B73 beta-carotene hydroxylase 1 (CrtR-B1) gene, complete cds. It is the same inbred from which the reference genome of maize is generated, hence they are one and the same (Lines 328-333).

Comment 5: L315: GQ889870 is a sequence with 1250 insertion. It is not clear where the information about the 325 bp insertion comes from.

Response 5: Thanks for suggesting critically. The gene ID, GQ889716 represents the sequence with insertion of both transposable elements (325 bp and 1250 bp), acts as an unfavorable sequence and the sequence with gene ID GQ889870 acts as a favorable sequence for the crtRB1 gene. The manuscript text has been revised as per the suggestion (Lines 328-333). 

Comment 6: L319: Li et al. have published primers labelled as InDel7. Why did you choose InDel118?

Response 6: We appreciate the reviewer’s suggestion. In the study highlighted by Li et al. (2012 https://doi.org/10.1371/journal.pone.0036807) on functional polymorphism for the vte4 gene, they reported two InDels, namely InDel7 and InDel118, both significantly associated with higher α-tocopherol. As the InDel size was larger in InDel118, we used InDel118 to avoid any other overlapping amplicons with better resolution in the multiplex assay (Lines 67-73, 333-337).    

Comment 7: L333: Please add a column with the primer names in the table. For example, Set 1 Forward = S1F. In this way, the names of the primers in Figure 2 are easy to understand. Also indicate which allele is amplified with Reverse-1 and which with Reverse-2. For each unfavourable allele, also indicate which primers are used to amplify it. What is the reason for obtaining unfavourable alleles of 171, 500, 1096 and 1425 bp? For example, for Set-3, only allele 213 bp is mentioned as unfavourable in the Results section, while also alleles 869, 1138 and 1749 bp are mentioned here.

Response 7: Thanks for suggesting critically. We have added a column to represent the primer names with different combinations of forward and reverse primers. Also, we provided the primer name in obtaining the particular amplicon size. The relevant text has been included in Table 4 as per the suggestion (Lines 278-279).

Comment 8: L3712: …reagents was increased. – …were increased.

Response 8: The grammatical error was rectified as per the suggestion.

Comment 9: L401: In the Discussion section it is mentioned that: “The multiplex- PCR assay developed here saved ~42% of the cost and ~36% of the time when compared with the total cost and time involved in two uniplex- PCR assays.” But here 50% of the time and 41%. Which information is correct?

Response 9: Thanks for the critical observations. The values have been crosschecked and corrected for uniformity across the sections as per the suggestion (Lines 24-25, 419-421).

Comment 10: Results: L85: You do not need to insert nucleotide sequences here if they are already contained in Table 3. How was set-3 selected? What does it mean that "it was compatible”? (I saw that this is explained in the Discussion, but it should be here.)

Response 10: The primer sequences have been removed from the text as per the suggestion.

Comment 11: L85: …with aother – …with another

Response 11: The typo is corrected as per the suggestion.

Comment 12: Discussion: L200-L211 – this should be included in the introduction.

Response 12: We appreciate the reviewer’s concern. The entire manuscript is based on the content discussed in this paragraph, it would be easier and clearer for the readers to better understand the findings of the current study.

Comment 13: L241: Please add in the title that the numbers are based on the sequence GQ889716. Also, I think Figure 2 should be included in the Results section as a primer map and a comparison of primer positions from previous articles could also be provided. Also the resolution of Figure 2 should be increased.

Response 13: Thanks for the suggestion. The title of Figure 2 has been revised to include the “Numbers in the illustration are based on the sequence of GQ889716” as per the suggestion (Lines 250-253).

Comment 14: L246: Please include this information in the introduction to justify why you designed new primers.

Response 14: The study mainly focused on developing the breeder-friendly multiplex assay for accelerating the maize breeding program, the sentence here is appropriate for the effective conveyance of our findings with better clarity.    

Comment 15: L247-L265: I think this should be under the Results section. Here you explain well why Set-3 was chosen, but this should be in the results. The Discussion section is where you compare the results to other studies, how your results improved on previous methods, how your methods differ, etc.

Response 15: We appreciate the reviewer’s concern. As the current study is a unique of its kind in developing the multiplex assay for the crtRB1 and vte4 genes, in this section we discussed the different possible combinations with appropriate reasons and also, we compared our findings with original studies as per the suggestion.  

Comment 16:  L269-L285: This paragraph could be shortened as the general introduction to the multiplex assay is already included in the Introduction section.

Response 16: Thank you very much for the suggestion. The paragraph has been to provide a crisp and clear message without overlapping with content in the earlier sections (Lines 281-299). 

Comment 17:  L292: Shouldn’t it be PMI-PV1 instead of PMI-PV5?

Response 17: It has been revised to PMI-PV1 as per the suggestion.

Comment 18:  L296: Why crtRB1/crtRB1/vte4+/vte4 is the most favourable genotype for further backcrossing? Why not homozygotes with vte4 (allele for high content of gamma tocopherol)?

Response 18: We agree that the selection of the favorable homozygote for vte4 would be more rewarding in the backcrossing program. Among the possible genotypic classes obtained for segregation of crtRB1 and vte4 in the BC1F1 generation screened in the study, the plant with allelic status of crtRB1/crtRB1/vte4+/vte4 is more rewarding as it is favorable homozygote for crtRB1 and favorable heterozygote for vte4 and none of the other genotypic classes are favorable homozygote for vte4 gene. Thus, the plants with genotypic crtRB1/crtRB1/vte4+/vte4 are the most favorable genotype. The relevant content was revised for better clarity as per the suggestion.

*Note: Line numbers provided in the response to each comment here are according to the track change copy of the manuscript.

Round 2

Reviewer 1 Report

Comments and Suggestions for Authors

The authors carefully revised the manuscript according to the review comments and did not find any obvious errors or problems.

Author Response

We sincerely thank the Academic Editor and the Reviewers for the constructive suggestions for improving the manuscript. All the suggestions have been incorporated in the manuscript and the revised manuscript is submitted for your kind consideration. Pointwise responses to each comment of the reviewers are submitted as separate files for each reviewer.

Response to reviewers’ comments

Comment: The authors carefully revised the manuscript according to the review comments and did not find any obvious errors or problems.

Response: We greatly appreciate your feedback and assistance in improving the quality and recognizing the importance of our work. Thank you again for your valuable contribution.

Reviewer 4 Report

Comments and Suggestions for Authors

The manuscript has been greatly improved. I have only few comments:

L65: This sentence: “The 3′TE polymorphism of the crtRB1 gene that spans the 6th  exon, the allele without TE insertion is the favourable allele responsible for enhancing the β-carotene [26, 30].” should be corrected. Maybe like this: “The 3′TE polymorphism (insertions of 325 bp or 1250 bp) of the crtRB1 gene that spans the 6th exon results in the unfavorable alleles, whereas the allele without TE insertion is the favourable allele responsible for enhancing the β-carotene [26, 30]. “? It is also possible that I didn’t understand correct meaning.

L81: Why did you include a paragraph about KASP? When KASP method is developed, it is not necessary to sequence the amplicons? Correct?

L112: Please correct "unfaourable" to unfavourable/unfavorable.

L216: "Allele with ‘+’ sign indicates the wild-type (unfavourable allele)" – but crtRB1 is wild type and it is favorable?

Author Response

We sincerely thank the Academic Editor and the Reviewers for the constructive suggestions for improving the manuscript. All the suggestions have been incorporated in the manuscript and the revised manuscript is submitted for your kind consideration. Pointwise responses to each comment of the reviewers are submitted as separate files for each reviewer.

 Response to reviewers’ comments

Reviewer #4:

Comment 1: L65: This sentence: “The 3′TE polymorphism of the crtRB1 gene that spans the 6th exon, the allele without TE insertion is the favourable allele responsible for enhancing the β-carotene [26, 30].” should be corrected. Maybe like this: “The 3′TE polymorphism (insertions of 325 bp or 1250 bp) of the crtRB1 gene that spans the 6th exon results in the unfavorable alleles, whereas the allele without TE insertion is the favourable allele responsible for enhancing the β-carotene [26, 30]. “? It is also possible that I didn’t understand correct meaning.

Response 1: We appreciate your feedback. The sentence suggested by you provides more clarity than the previous one, we have accepted the sentence correction as follows: "The 3′TE polymorphism (insertions of 325 bp or 1250 bp) of the crtRB1 gene that spans the 6th exon results in the unfavorable alleles, whereas the allele without TE insertion is the favorable allele responsible for enhancing the β-carotene" (Lines 64-69).

Comment 2: Why did you include a paragraph about KASP? When KASP method is developed, it is not necessary to sequence the amplicons? Correct?

Response 2: Thank you greatly for suggesting critically. The paragraph related to KASP was included based on a suggestion from one of the reviewers during the previous round of the review process. Yes, we agree that once the method is developed in KASP, there is no need to sequence the amplicons. The paragraph has been revised for more clarity of the content and message, as per the suggestion (Lines 83-89).

Comment 3: L112: Please correct "unfaourable" to unfavourable/unfavorable

Response 3: Thanks for your critical observations. The typo has been corrected as per the suggestion (Line 111).

Comment 4: Allele with ‘+’ sign indicates the wild-type (unfavourable allele)" – but crtRB1 is wild type and it is favorable?

Response 3: Thanks for the comment. The allele with ‘+’ sign indicates the wild-type (unfavourable allele)" and the allele without any sign (crtRB1) indicates the mutant type (favourable allele). These points have been modified accordingly to the text (Lines 258-259).  

*Note: Lines numbers provided in the response to each comment here are according to the

track change copy of the manuscript.
